# Exploring the Perceptions of Undergraduate Pharmacy Students’ Communication Skills to Facilitate Better Professional Decision-Making in the UK

**DOI:** 10.3390/pharmacy13050117

**Published:** 2025-08-28

**Authors:** Aadesh Dave, Sukvinder Kaur Bhamra

**Affiliations:** Medway School of Pharmacy, University of Kent, Chatham, Kent ME4 4TB, UK; a.dave@kent.ac.uk

**Keywords:** communication, pharmacy, pharmacist, education, professional decision-making, interprofessional education, simulation

## Abstract

**Background**: Pharmacy professionals have an important role in delivering patient-centred care, with effective communication skills forming the foundation of interactions with patients. This study aimed to explore how pharmacy students perceived their own communication skills; along with the communication skills education and training experiences in their undergraduate pharmacy degree in the UK. **Methods**: A 22-item questionnaire was designed and piloted before being distributed online. Snowball sampling was employed to recruit participants undertaking an undergraduate pharmacy degree. Quantitative statistical and qualitative thematic analysis was conducted. **Results**: A range of pharmacy schools were represented in the data set (n = 10) with 217 responses collected. Participants rated their communication skills highly (53.03%, n = 114), but stated they still required improvement (79.72%, n = 173). A proportion of participants stated that they could appropriately make professional decisions (52.08%, n = 100) and that their communication skills had facilitated their professional decision-making skills (57.89%, n = 110). Effective teaching methods reported included role play with peers (80%, n = 156) and small-group teaching sessions (64.10%, n = 125). Participants felt that interprofessional education and simulated patients could help improve their communication skills further. **Conclusions**: Communication education is a crucial element in developing future healthcare professionals. Thus, investment in resources is required to facilitate communication skills in the earlier stages of the undergraduate pharmacy degree.

## 1. Introduction

### 1.1. Importance of Effective Communication in Healthcare

Pharmacy professionals have an important role in the delivery of person-centred care, with effective communication and the consultation processes forming the foundation of interactions with patients. Effective communication skills allow for a positive patient–healthcare professional (HCP) rapport to be built, which in turn can lead to patient satisfaction and better health outcomes [1,2,3]. Studies have consistently shown that interaction with healthcare professionals, such as pharmacists, significantly improves medication adherence. A study found that pharmacist-led interventions, such as medication reviews and counselling, resulted in a 20% increase in adherence rates among patients with chronic conditions [4]. These findings underscore the critical role pharmacists have in enhancing medication adherence through direct patient interaction. The ability of pharmacy professionals to demonstrate active listening, empathy and clear communication is important during clinical consultations. With the evolving role of pharmacy professionals in the UK (i.e., pharmacists as independent prescribers) and the developments in virtual consultations, there is a necessity for pharmacy students to have high-quality communication skills to facilitate valuable consultations. Effective communication skills are also instrumental when liaising with other HCPs across all sectors of healthcare.

### 1.2. International Relevance of Communication Skills

Globally, the International Pharmaceutical Students’ Federation (IPSF) and the International Pharmaceutical Federation (FIP) have highlighted the importance of communication skills as core competencies for pharmacy professionals [5,6]. They have developed educational tools, targeted towards pharmacy students and pharmacy schools, to enhance communication learning techniques. Some of these educational tools include simulation-based learning sessions involving role-playing scenarios and patient interactions to provide practical communication experience within a controlled environment. Online modules and e-learning platforms have also provided flexible, self-paced learning opportunities, covering topics like patient counselling, cultural competence and handling difficult conversations [5,6]. Historically, Kurtz and Silverman defined communication skills through the practical teaching tool, the Calgary–Cambridge Guide [7]. The guide emphasises the development of effective communication skills by focusing on core functions such as building rapport, active listening and gathering accurate information from the patient. The importance of incorporating shared decision-making in this process is further underlined. The consultation model delineates and structures the skills to aid patient–healthcare professional communication. Primarily aimed at medical students, many pharmacy schools teach models like this as part of their curriculum. These tools have been used as part of a broader effort to ensure that all pharmacy professionals are equipped to engage effectively with patients and other healthcare providers, ultimately to improve patient care and outcomes through better communication.

### 1.3. Pharmacy Education in the UK

The General Pharmaceutical Council (GPhC), the regulatory body for pharmacy in the United Kingdom (UK), is responsible for setting standards for pharmacy professionals and undergraduates. Part of these standards emphasise how effective communication is a priority to deliver safe and effective care [8]. Specifically, effective communication is essential to the delivery of person-centred care and to working in partnership with others [8,9]. Pharmacy schools in the UK offer a four-year undergraduate Master of Pharmacy (MPharm) degree, which is accredited by the GPhC. On completion of the MPharm degree, pharmacy students undertake a year of foundation training before they can register as a pharmacist. Communication skills are integral to the MPharm degree, and pharmacy schools play a key role in integrating these skills when shaping future pharmacists [9]. Changes in UK pharmacy education standards have focused on increasing the clinical aspects of pharmacy practice, including communication skills. All pharmacy graduates must now prepare to be independent prescribers, but it is not known how this additional requirement has impacted pharmacy students and pharmacists [10,11,12,13]. Furthermore, pharmacists and employers have already reported a need for an increased focus on clinical training in the pharmacy curriculum, including more training in patient communication skills prior to this [14,15,16]. Clearly, effective communication skills are crucial to the development of pharmacists, and pharmacy students can provide valuable feedback to educators and curriculum planners to improve communication skills training (e.g., via course evaluations) within their curriculum. However, there is little understanding of UK pharmacy students’ perceptions, specifically, of their communication skills to facilitate better professional decision-making. As there are so many different learning and teaching methods for communication skills training, exploring the type of learning integrated into UK undergraduate pharmacy curricula will be vital for informing how improvements can be made.

### 1.4. Research Study Rationale

While research has been performed to evaluate the communication skills of graduates and practising healthcare professionals, there is very limited work that has explored the perceptions of undergraduate pharmacy students [17,18] in the UK. Investigating how communication skills are taught within undergraduate curricula will help to identify potential education, training and learning needs to improve current communication training. Furthermore, the competence and barriers faced in translating theoretical knowledge into professional decision-making. Therefore, the aim of this study was to explore how pharmacy students perceive their own communication skills and training experiences during their undergraduate pharmacy degree using a mixed-method approach. In addition, this study sought to uncover the different communication skills training provided in UK pharmacy schools, offering a perspective that is crucial for refining pedagogical approaches in UK pharmacy education.

## 2. Materials and Methods

### 2.1. Initial Phase Activities

A questionnaire in Appendix A was developed to investigate UK-based pharmacy students’ opinions of their own communication skills education and training experiences. The questionnaire was formulated after a narrative review was conducted by the research team and incorporated their own experiences teaching communication skills at higher education institutes. The initial draft of the questionnaire was reviewed by the research team, followed by the ethics committee, before a pilot study was conducted. The pilot study was conducted with five prospective participants. Cognitive interviews formed part of the pilot study to ensure the questionnaire was clear and unambiguous. Feedback on the structure, content, wording, instructions, layout and overall time burden was considered before minor amendments were made to the final version of the questionnaire. The amended version of the questionnaire was resent to the ethics panel for final approval before distribution via an online software, SurveyMonkey^®^ (San Mateo, CA, USA) began. The Internet Protocol (IP) addresses were restricted by SurveyMonkey^®^ to prevent multiple responses being submitted by the same individuals.

### 2.2. Study Design

A mixed-methods questionnaire utilising both quantitative and qualitative questions was used. The questionnaire was composed of a range of multiple-choice, Likert-scale and free-text questions. Key topics included personal evaluation of communication skills, review of educational and training activities undertaken and consideration of impact on professional decision-making skills. As this was a cross-sectional feasibility study, the number of participants to be recruited was not pre-determined. An approximate target sample size of 225–450 participants was determined by aiming to recruit 5–10% of the estimated eligible population, as per the Universities and Colleges Admissions Service (UCAS) records, where approximately 4400 pharmacy students were enrolled in 2021.

### 2.3. Participant Recruitment

The online questionnaire was shared by the research team via personal and professional networks and social media (i.e., LinkedIn, Twitter/X, WhatsApp, Instagram and Facebook). The research team actively shared the questionnaire to raise awareness of the study, thus leading to an overall snowball sampling technique being employed. The questionnaire was available online for three months, from October to December 2022. This allowed sufficient time for the questionnaire to be circulated online and reminder notifications to be posted. Participant inclusion criteria: participants were required to be over 18 years of age and currently undertaking the MPharm programme in the UK. Participant exclusion criteria: participants under 18 years of age, not undertaking the MPharm programme in the UK.

### 2.4. Informed Consent

Before participants decided to complete the questionnaire, they had access to the participant information leaflet S1 to read about the project prior to consenting to take part. The consent procedure included asking questions about eligibility to ensure participants met the inclusion criteria. All responses were further screened by the research team to ensure participants met the inclusion criteria. No personally identifiable data was obtained to ensure anonymity.

### 2.5. Ethics Approval

Ethics approval was granted from the Medway School of Pharmacy at the University of Kent (REF:021022).

### 2.6. Data Analysis

Data analysis methods included using the Statistical Package for the Social Sciences (SPSS) software, version 27, to input the quantitative data from the questionnaires and analyse the results through descriptive statistics (i.e., frequencies) and inferential statistics (i.e., analysis of variances). This enabled descriptions, comparisons and calculations from key features of the data set to be made. This includes means, variances and confidence intervals. NVivo software, version 14, was used for thematic analysis of qualitative data from the questionnaire. This included themes, topics and trends from open-ended responses. Each potential theme was clearly defined, and data segments associated with each theme were reviewed to ensure internal coherence and external heterogeneity. Microsoft Excel 2010 was used to create graphical representations. All data analysis was undertaken, systematically discussed and reviewed by the research team to confirm the results. For questions which were not completed by all participants, n = x was used to represent the response rate. Participant responses presented in the results were coded with a number (e.g., P001), gender (i.e., M-male, F-female) and year of study (ranging from 0 to 4).

## 3. Results

### 3.1. Demographics

A total of 217 participants completed this study, representing 10 pharmacy schools across the UK. There was a higher proportion of females (74.7%, n = 162), aged 18–20 years (n = 114), forming the study population (Table 1). Participants represented a diverse range of ethnicities, and a large number of participants stated that English was their first language (71.4%, n = 155). A large proportion of participants were undertaking the 4-year undergraduate degree (65.4%, n = 142), with a small proportion of participants undertaking the 5-year MPharm degree, which incorporates preparatory year training (30%, n = 65).

### 3.2. Self-Assessment of Communication Skills

Participants were asked to rate their communication skills from a scale of one (poor) to five (excellent). The results showed that over half of participants rated their communication skills highly as good or excellent (53.03%, n = 114), although some participants still felt their communication skills were average by scoring it at three (41.86%, n = 90). Despite this, a large proportion of participants indicated that their communication skills still required improvement (79.72%, n = 173). There was no significant difference in self-reported rating of communication skills across the different stages of the pharmacy degree. Furthermore, there was no significant difference in responses from participants based on demographics, including whether their first language was English or not. It was noted that communication skills were taught in all years of the undergraduate degree, but most participants recalled it being definably taught in the early years of the respective undergraduate degree in year one (73.06%, n = 141) and year two (60.62%, n = 117). Some participants did not think or were unsure whether they had the opportunity to undertake interprofessional education as part of their training (67.71%, n = 130).

Participants were asked to rank several statements about their communication skills (Figure 1). Nearly half of the participants agreed that they received theoretical communication skills training in their undergraduate degree (47.69%, n = 103). This was also the case when comparing whether participants received practical communication skills training in their undergraduate degree (51.63%, n = 111). It was noted that a small proportion of participants disagreed that they had received feedback on their communication skills (28.84%, n = 62).

A series of one-way Analysis of Variance (ANOVA) tests were conducted to examine the effect of the year of study on different aspects of communication skills training and feedback received during the undergraduate pharmacy degree. For the dependent variable of having received theoretical training, there was a statistically significant effect of the year of study, F(4211) = 5.882, *p* < 0.001. Regarding practical training, a statistically significant effect of the year of study was also found, F(4210) = 2.457, *p* = 0.047. Furthermore, for having received feedback on communication skills, the analysis revealed a statistically significant effect of the year of study, F(4210) = 5.155, *p* < 0.001. Following this, Tukey’s Honestly Significant Difference (HSD) post hoc tests were conducted to explore specific differences in perceived communication skills training and feedback across different years of study in pharmacy programmes. The results indicated several statistically significant differences. Specifically, students in Year 3 reported significantly higher levels of theoretical communication skills training compared to students in Year 0 (M = 0.61, *p* = 0.048, 95% CI [0.00, 1.22]), those in Year 1 (M = 0.93, *p* < 0.001, 95% CI [0.38, 1.47]) and those in Year 2 (M = 0.71, *p* = 0.008, 95% CI [0.13, 1.29]). Year 3 students further reported significantly higher levels of practical communication skills training compared to students in Year 2 (M = 0.66, *p* = 0.019, 95% CI [0.07, 1.24]. In addition, Year 3 students reported significantly higher levels of communication skills feedback compared to students in Year 0 (M = 0.71, *p* = 0.037, 95% CI [0.03, 1.14]), Year 1 (M = 0.72, *p* = 0.011, 95% CI [0.11, 1.33]) and Year 2 (M = 1.03, *p* < 0.001, 95% CI [0.39, 1.69]). No other pairwise comparisons reached statistical significance.

A proportion of participants indicated that they were able to counsel a patient on their medication (38.89%, n = 84). Approximately half of the participants did not agree that they could discuss concerns about an incorrect dose of a medication with a healthcare professional and subsequently provide a suitable recommendation (50.01%, n = 108). Some participants stated that their communication skills had facilitated their professional decision-making skills (68.42%, n = 130). A small number of participants responded to the statement about interprofessional education (29.62%, n = 64), and few participants stated that they strongly agreed that interprofessional education improved their communication skills (3.13%, n = 2).

### 3.3. Communication Skills Teaching Methods

Participants were asked to rank common communication skills teaching methods based on their perceived effectiveness (Figure 2). A large proportion of participants (80%, n = 156) stated they undertook role play with peers as part of their undergraduate degree, although a proportion stated that this was not perceived as the most effective teaching method (69.74%, n = 53). Reflective writing (48.15%, n = 13) and online learning materials (52.94%, n = 9) were perceived as the least effective teaching methods, with practice with real patients (46.97%, n = 62) being perceived as the most effective. Placements were perceived as not effective for some participants (21.88%, n = 14), along with observations from teachers (39.13%, n = 9). A third of the participants revealed problem-based learning (32.82%, n = 64), and a quarter of participants suggested team-based learning (25.13%, n = 49), as teaching methods used for communication skills training. However, some participants perceived that both problem-based learning (65%, n = 13) and team-based learning (73.68%, n = 14) are not the most effective.

### 3.4. Qualitative Review of Communication Skills

Thematic analysis of qualitative free-form open question responses (41.67%, n = 90) uncovered key themes about how communication skills could be improved. This included increasing exposure to patients as part of education and training (Figure 3).

Some participants stated that communication skills could be greatly improved through more patient contact. A participant stated, “scenario-based training would be helpful as it mimics real life“ [P066:F4], and another participant said, “providing patient education with simulated patients would help” [P113:M3]. Small group sessions with peers were identified as a facilitator of improving communication skills: “smaller groups are easy to communicate with” [P100:F0]. This was echoed by others who felt more confident speaking in smaller groups. Interprofessional education was found to be a potential answer to the lack of access to other healthcare professionals by enabling interaction with students from other healthcare professions as part of communication skills training: “presenting ideas and findings to healthcare professionals would help” [P009:F3] and “more practice communicating with other healthcare professionals” [P158:M3]. On a positive note, designated communication skills sessions were highlighted as a facilitator of communication skills training, with participants perceiving this teaching method as effective in improving their confidence. However, language and cultural differences were identified as potential barriers to effective communication.

Facilitators were deemed to exist within the environment and aimed to aid the process. An improvement represented a suggestion for future enhancement or a recognised positive change from a previous state for the future. Facilitators describe ”what works now”, while improvements describe “what could work better” or “what has recently changed for the better”. During coding, a member of the team consistently applied these definitions, asking whether the data segment referred to an existing supportive element or a proposed or actual change aimed at enhancement. This allowed us to reliably separate these concepts while analysing participants’ perspectives.

## 4. Discussion

### 4.1. Implications of Pharmacy Education and Practice

This study examined the perspectives of UK undergraduate pharmacy students’ communication skills during their undergraduate pharmacy education. The key findings showed that students from the pharmacy schools, who responded to the questionnaire, tend to experience positive attitudes toward their varied communication skills education and training. This was regardless of gender, ethnicity and whether English was their first language or not.

Our findings further suggest that students had less confidence in navigating complex conversations. This extends to a similar theory–practice gap reported in a study exploring UK pharmacy professionals’ communication skills, which uncovered a lack of undergraduate training [9]. In contrast, our study provides novel insights by revealing that this gap was perceived to stem from suboptimal exposure to patients and interprofessional learning opportunities. If further exposure to this training can be incorporated, this can bridge the gap between undergraduates and professionals in practice [9].

The findings of this study highlight the importance of teaching communication skills in the pharmacy curricula. Furthermore, the need to incorporate strategies for building student resilience and confidence in managing more challenging patient interactions. Therefore, we recommend the integration of more high-fidelity simulation-based learning, with other healthcare professions, focused on affective and ethical challenges. This could better prepare students for the complexities of professional decision-making and align with the GPhC’s emphasis on person-centred care [8,10].

### 4.2. Interactive Engagement

The study showed that interactive practice with peers, through role play, created positive experiences that built student confidence and ensured satisfaction. Particularly, in the earlier years of the undergraduate pharmacy degree, where theoretical and practical training are provided most intensely. The development of key communication and professional decision-making skills has been seen in earlier years of healthcare programmes, such as pharmacy [19,20,21]. These skills are required for professional success. The skills implement theoretical learning and build upon practical experiences to enable early integration and ensure better preparedness. This is in line with the results of this study, which indicated the perceived level of theoretical and practical training, along with communication skills feedback, varied significantly depending on students’ year of study. The findings also suggest that the perceived theoretical or practical training and feedback in relation to communication skills significantly increases by at least the third year of the pharmacy programme.

Designated small-group communication skills sessions, involving role play with peers, were found to facilitate effective communication skills. Small group sessions have been found to be effective in improving communication skills through building confidence and learning from colleagues [22,23]. One study found that smaller groups encouraged participation from students who were reserved and allowed more personalised feedback opportunities [22]. However, the standardisation of this learning was questioned, with poor practices potentially being acquired from peers. Role play was found to be most useful, as it allows the practice of communication in a safe, simulated and controlled environment. Whilst further providing opportunities for learners receive formative feedback [24]. This was highlighted as an issue in this study, as students felt they had not received enough feedback. Previous studies have shown that role play is effective in enhancing communication skills [22] and that students consider role play as an essential tool to acquire effective communication skills. In one study, almost all students reported immense confidence in communicating medication details, and the majority reported better retention of pharmacological concepts [25]. Therefore, students have continued to advocate for the sustained inclusion of small-group role play with peers in the curriculum as a valuable teaching and assessment tool. It is important to note that there could potentially be a lack of realism associated with peer role play [26], and standardised patients providing professional feedback could benefit the training of communication skills further [27]. There are limited comparative studies examining the effectiveness of peer, patient and healthcare professional small-group teaching available in the literature [28]. Furthermore, the positive outcomes from the inclusion of patients and healthcare professionals in communication skills training within undergraduate pharmacy education are generally unknown.

### 4.3. Patients in Teaching

To obtain positive outcomes from undergraduate healthcare professionals’ curriculum, it is important to have sufficient and varied education and training [28]. The results of this study highlight that improvements can be made. There is a call for more practice with real patients, as students expressed a desire for more practical training with real patients in addition to simulated patients. Studies have found that patient contact in healthcare programmes, such as pharmacy, is a vital part of developing student communication skills. It is suggested that patients provide learners with practical experiences within their discipline through exposure to real-life complexities, which cannot be replicated through simulated environments [29,30,31]. These opportunities enable students to truly reflect on their experiences and obtain appropriate feedback for their learning [29]. This explains the positive correlation seen in the study, as students who stated that they received practical training were also confident to introduce themselves to their patients and counsel patients on their medication.

The limited or lack of access to real patients was identified as an issue by participants in this study. A recent study highlighted a significant decline in patient engagement in volunteering post-pandemic [32]. General studies on volunteerism post-pandemic reported that universities have seen a drop in volunteer numbers, along with the amount of time volunteers are willing to provide support [33]. This decline reflects a reduced willingness and broader hesitation among patients and volunteers alike in the post-pandemic era. Significant pressures within the healthcare system in the UK are also a barrier to accessing patients for teaching. A recent study found that the pressures on the NHS, have made it increasingly difficult to provide adequate clinical placements and patient contact opportunities for students undertaking healthcare degrees [34,35]. This has been due to staffing shortages, patient demand and financial constraints. These challenges have led to limited supervision and reduced hands-on experience, which are crucial for the professional development of students. With the ongoing financial challenges in higher education, funding for patients in teaching is less likely to be considered a priority. There are also ethical implications involving patients in teaching that need to be considered [36,37]. Simulated patients, such as professional actors, could be utilised to mitigate this challenge. One study involved an actor-led forum theatre, where students redirected scenes depicting pharmacist-patient consultations. Students learned about their own communication styles and role-played consultation situations with professional actors. Outcomes were positive as students felt the session was useful in the development of communication skills [22].

### 4.4. Interprofessional Education

Interprofessional education with students from other healthcare professions was suggested as a solution to the lack of access to healthcare professionals for communication skills training [38,39,40,41]. Interprofessional education was not deemed a facilitator for developing communication skills by participants in this study. In fact, observing healthcare professionals was perceived one of the least effective teaching methods, with most participants unsure of its effectiveness. Still, it is deemed as a vital part of developing communication skills in pharmacy students as it follows a multidisciplinary approach to teaching [38,39]. This is further in line with the new requirements for the initial education and training of pharmacists set out by the regulator [40]. Studies typically describe pharmacists’ interprofessional encounters as allowing them to learn about other health professionals and build collaborative relationships [39,40,41,42,43]. They suggest a framework for pharmacy and medical school training to move from solitary educational experiences to synergistic learning opportunities to develop key clinical skills and confidence to communicate in a team environment [41,42]. Qualitative analysis revealed that students perceived that they had learnt about different scope of practices through this and built confidence in their communication skills [43]. Although, the skills of the facilitator and preparation for the experience were perceived to promote the success of these events.

The strain on healthcare resources often means that students receive less hands-on training and supervision, which is essential for their professional development [34,35]. These pressures may result in limited time and resources for mentorship and supervision. This may lead to reduced opportunities for students to engage in healthcare contact and clinical learning. Interprofessional education opportunities, integrated through the curricula, can help overcome the current pressures surrounding the availability of healthcare professionals. These opportunities may further harness collaboration between different healthcare schools. Though financial and logistical investments are required to facilitate these large-scale collaborations between healthcare disciplines, they are important to ensure the quality of learning and can significantly influence students’ experiences [44].

### 4.5. Strengths, Limitations and Future Research

This study adds to and updates the limited evidence from undergraduate students about the perceptions of communication skills education and training provided in pharmacy schools within the UK. The use of open and closed questions in this study was highly valuable, as the level of detail participants disclosed may not have otherwise been achieved. As the sample size was small, the results are not representative or generalisable on a larger scale. The use of snowball sampling through social media, while effective in reaching a broad audience quickly, may have introduced a sampling bias. This could potentially skew the findings toward individuals who are more digitally connected or interested in the topic. The progressive decline in questionnaire completion might have been attributed to the length of the survey. The cross-sectional design of the study meant that no causal inferences could be drawn. Here, longitudinal studies are needed to study the direction of association between participant attitudes and training provided, and to follow up on different years of study.

In future studies, including both students and educators in the sample would make it possible to explore the associations between: different types of training provided, individual perceptions and suggested improvements. In addition, focus group discussions should be conducted to allow a more comprehensive discussion to take place, alongside open questions answered by participants in the questionnaire. This will allow for a thorough exploration of the perceptions of communication skills education and training in undergraduate courses. Lastly, targeting graduates and early career pharmacists (during and after their first year of practice) would be beneficial to further explore how communication skills develop with time and experience.

## 5. Conclusions

This study indicates that there is an association between having more peer-, patient- and healthcare-focused communication training, and a positive influence on students’ self-reported communication skills. This, in turn, facilitates better professional decision-making. Furthermore, a programme design that integrates communication skills, via small group sessions, may prepare students better. Some suggestions for improvements include: more training with patient volunteers through a variety of placements, interprofessional education with students of other healthcare professions and additional formative opportunities to practice and obtain personalised feedback.

Overall, this study has provided a unique insight into how undergraduate pharmacy students perceive their communication skills based on the experiences from their undergraduate degree. The key findings can be taken forward to better support preparation and refinement for accreditations of the undergraduate pharmacy degree. Forthcoming work should focus on targeting graduates and early career pharmacists, along with in-depth analysis via focus groups. This would support additional changes to the curriculum to be recommended. Resources and investments will also be required to support these recommendations and establish the foundations of communication skills in the earlier stages of the undergraduate pharmacy degree.

## Figures and Tables

**Figure 1 pharmacy-13-00117-f001:**
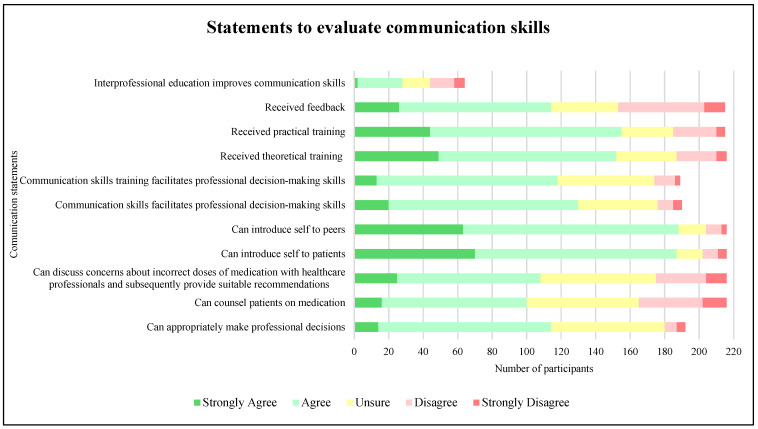
Participant responses to ranking statements about communication skills (n = 64–216).

**Figure 2 pharmacy-13-00117-f002:**
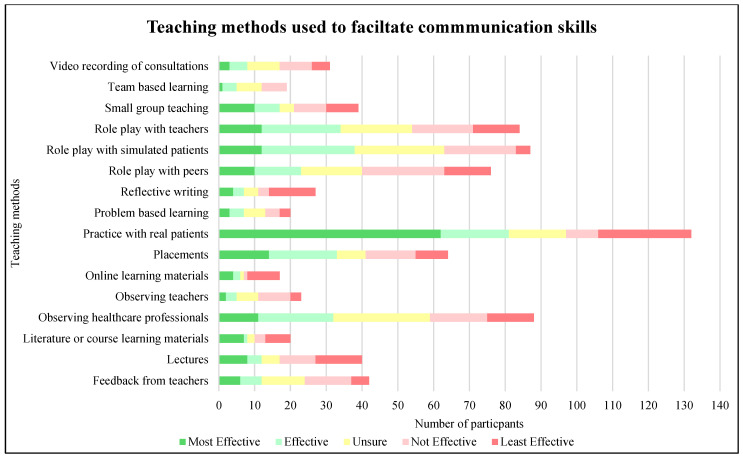
Participant responses to ranking communication skills teaching methods (n = 17–132).

**Figure 3 pharmacy-13-00117-f003:**
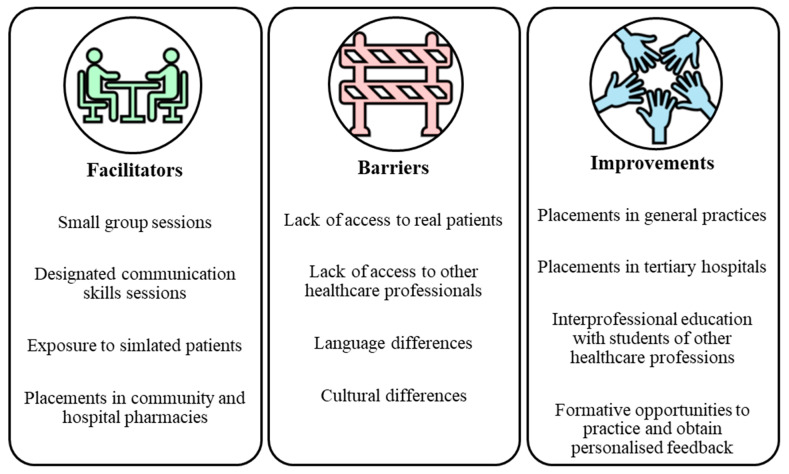
Thematic analysis of qualitative responses uncovering key facilitators, barriers and improvements required to develop communication skills.

**Table 1 pharmacy-13-00117-t001:** Participant demographics.

Demographics (n = 217)	Frequency (n)	Percent (%)
Gender		
Female	162	74.7
Male	54	24.9
Non-Binary	1	0.5
Age		
18–20 years	114	52.5
21–30 years	91	41.9
31–40 years	9	4.1
41–50 years	3	1.4
Ethnicity		
White: British	16	7.4
White: Irish	2	0.9
White: Other	7	3.2
Black: African/Caribbean	77	35.5
Black: Other	2	0.9
Arab	8	3.7
Asian: Indian	30	13.8
Asian: Pakistani	17	7.8
Asian: Bangladeshi	20	9.2
Asian: Sri Lankan	4	1.8
Asian: Chinese	3	1.4
Asian: Other	21	9.7
Mixed Race	5	2.3
Iranian	2	0.9
Kurdish	3	1.4
English as their first language		
Yes, English was their first language	155	71.4
No, English was not their first language	62	28.6
Type of MPharm degree registered on		
4-Year MPharm	142	65.4
5-Year MPharm	65	30.0
MPharm Extended	10	4.6
Current year of study		
Year 0 (Preparatory Year)	36	16.6
Year 1	58	26.7
Year 2	43	19.8
Year 3	41	18.9
Year 4	39	18.0

## Data Availability

The data that support the findings of this study are available from the corresponding author upon reasonable request.

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
