# Peer review of "Exploring the Perceptions of Undergraduate Pharmacy Students’ Communication Skills to Facilitate Better Professional Decision-Making in the UK"

_pharmacy, 2025, doi:10.3390/pharmacy13050117_

Round 1
Reviewer 1 Report
Comments and Suggestions for Authors
Comments for Author
This is an interesting, well-written paper. I did however note a few significant issues with this paper, per below:
Methods: The three primary aims of this study were to determine how pharmacy students in the UK perceive their communication skills training experiences during their undergraduate curriculum; to discover the types of communications methods used in their curricula and their associated merits; and to evaluate the students’ perceptions of the impact of their communication skills preparation on their ability to execute professional decision-making skills. In my mind, the optimal cohort of students to survey to address the study aims would be students who recently completed the pharmacy curriculum at a UK college of pharmacy (i.e., completed the 4-year program, as stated in line 54). Even better, a cohort of graduates who recently completed the mandatory 1-year postgraduate training in the field would be optimally versed to address the study aims. It makes little sense to me to survey students who have not completed the bulk or all the curriculum, especially when your aims are to determine how well the curriculum prepared students to apply their communication skills. A Year 1 student, surveyed in October-December of their 1st semester, would only have a partial semester of the 8-semester curriculum to form their opinion, which in my view would be too limited to contribute to the study aims. Possibly the Year 4 student data could be analyzed, but the study N would be limited to less than 20% of the respondents (39 students), which may be a suboptimal sampling. It is difficult for me to know what an optimal sample size would be; adding the possible number of students/graduates/number of pharmacy schools in the UK would be helpful to frame this issue.
Results: include which school (can de-identify them if you wish), so the reader knows if the data came from one school, or numerous.
Discussion: study results indicate that there is room for improvement, calling for more experience with real patients, and a desire for interprofessional encounters within the 4-year required curriculum. While this is a meaningful finding for the UK to inform their curricular decisions, this would not be of widespread interest for the United States readership (and possibly other countries), as in the US, the required pharmacy curriculum for years have included both introductory and advanced pharmacy practice experiences (experiential learning in varied settings, placed early (introductory) and during the entirety of year 4 (advanced), which represents more than 1/3 of the 4-year required curriculum. The inclusion of Interprofessional Education in both didactic and experiential curricular components is also a required standard in US schools. Thus, the findings/plans for improvement have limited meaning/applicability to a broad audience.
Author Response
Dear Reviewer 1,
Thank you for your consideration of this article and your timely feedback. We really appreciate your comments and suggestions. We have taken this opportunity to address your comments. Please see the attached document with detailed responses to your comments.
On behalf of all the authors,
The Corresponding Authors

Reviewer 2 Report
Comments and Suggestions for Authors
Thank you for the opportunity to review this paper. This is an interesting paper on an important topic and I enjoyed reading it.
I offer the following comments to aid revision of the paper:
Introduction:
- Provides a thorough overview of existing research in this area and makes a strong case for this study. However, I found this difficult to read and didn't feel that information flowed logically between sections. Please consider reorganising information to improve the readability.
Methods:
- You mention that the survey was based on a literature review. Has this been published? If so, please include a citation as this will be of interest to readers.
- Did you think about targeting early career pharmacists/graduates in their first year of practice to explore how they view their training and development of their communication skills now they are in practice and are using these in patient care?
- How did participants provide consent and how were they screened to see that they met the inclusion criteria before completing the survey? E.g. was this through the survey itself or contact with the research team?
- Please provide more detail on the quantitative analysis that was performed (e.g. descriptive statistics)
Results:
- Is it possible to calculate an approximate response rate based on the total number of pharmacy students in the UK?
- You mention there was no significant difference in whether participants noted that their communication skills required improvement based on whether they spoke English as a first language. Did you look for and were you able to detect any differences based on demographics (e.g. age, gender or year of study)?
- In some parts you state that XX was the least effective method of teaching. You didn't objectively measure effectiveness, just participants perceptions of this so please make clear that these are references to perceived effectiveness.
- I am not sure what the sentence beginning on line 217 means- what is the information after the semi-colon referring to?
- The information in paragraph beginning line 250 is describing how you completed the analysis not your results—please consider moving this to the methods section
Discussion:
- Provides a thorough discussion of the results and how they fit into the broader literature and identifies key implications for policy and practice.
- There are some long sentences e.g. that beginning in line 299- please consider splitting these into shorter sentences to improve readability.
References:
- Were appropriate and support the content in the paper well. However, please check that these are all complete. Some references in your reference list (such as reference 4) are missing key details such as volume, issue and page numbers and doi.
Written expression:
- The paper is generally understandable but there are some minor spelling and grammar errors- please double check these. I have also flagged issues such as long sentences and organisation of information above.
I look forward to reading the next version of this paper!
Author Response
Dear Reviewer 2,
Thank you for your consideration of this article and your timely feedback. We really appreciate your comments and suggestions. Please see the attached document with detailed responses to your comments.
On behalf of all the authors,
The Corresponding Authors

Reviewer 3 Report
Comments and Suggestions for Authors
Thank you for the opportunity to review this manuscript. Overall, I found the article to be clearly written and methodologically sound, with conclusions that are well-supported by the data. The topic of communication skills training in pharmacy education is important, particularly in the context of patient-centred care. However, I believe there are some areas requiring clarification or refinement before the manuscript can be considered for publication. Additionally, while the study is competently executed, the potential impact and novelty of the findings appear somewhat limited, and the authors may wish to reflect further on the contribution of their work to the wider field.
Major Comments:
- While the topic is certainly relevant to pharmacy education, the originality and broader impact of the study may be considered modest. The authors are encouraged to better articulate how their findings extend or contrast with existing literature, and what specific gaps this study fills. A more critical reflection in the discussion section could enhance the manuscript’s value.
- Reference to Independent Prescribing (line 45) - The reference to pharmacists as independent prescribers must be contextualised. This is not a universal attribute of the profession and only applies in select countries (e.g., UK, parts of Canada, New Zealand). The statement should be revised or removed, or the countries where this applies should be explicitly named to avoid misleading generalisations.
- Scoping Review for Questionnaire Development - The authors mention conducting a scoping review to inform questionnaire design, which is commendable. However, no methodological details are provided. Given that scoping reviews are systematic by nature, the authors should briefly outline the search strategy, inclusion/exclusion criteria, and data sources used. Even a concise summary would significantly strengthen the transparency of the methods.
- Sampling Method and Bias Acknowledgment - The use of snowball sampling via social media and messaging platforms such as WhatsApp raises valid concerns about sampling bias and representativeness. This should be explicitly acknowledged in the limitations section, with reflection on how it may have impacted the findings.
- Sample Size Justification - Was any a priori sample size estimation performed to determine the minimum number of participants required to ensure representativeness or statistical power? Even in exploratory studies, a brief justification of the final sample size would improve methodological clarity.
- SPSS Version Disclosure - Please specify the version of SPSS used for the analysis, as this is standard in reporting quantitative data analysis.
- Inconsistent Wording in Results - The statement that “Participants agreed that they were able to counsel a patient on their medication” implies a general consensus, yet the numerical data provided (line 18–19) does not fully support this interpretation. The sentence should be rephrased to more accurately reflect the data, e.g., "A proportion of participants indicated..."
Minor Comments:
- Line 116: The article “a” before "higher education institutes" appears to be superfluous and should be removed.
Conclusion:
This manuscript presents useful insights into pharmacy students’ perceptions of communication training and their perceived competencies. With the revisions noted above — particularly the clarification of methodological aspects and contextual accuracy — the manuscript would be strengthened significantly. I look forward to reviewing a revised version.
Author Response
Dear Reviewer,
Thank you for your consideration of this article and your timely feedback.
We really appreciate your comments and suggestions. Please see the attached document with detailed responses to your comments.
On behalf of all the authors,
The Corresponding Authors

Round 2
Reviewer 1 Report
Comments and Suggestions for Authors
See attached file, below.

Author Response
Dear Reviewer,
Thank you for your consideration of this article and your timely feedback.
Please see the attached document with detailed responses to your comments.
On behalf of all the authors,
The Corresponding Authors

Reviewer 3 Report
Comments and Suggestions for Authors
The revised version submitted by the authors has satisfactorily addressed all the requested modifications I previously indicated as a reviewer, and I consider the manuscript suitable for publication.